# Knowledge, attitudes and practices (KAP) on COVID-19 among the general population in most affected districts of Nepal

Salina Shrestha[1], Rabin Malla [1]*, Sadhana Shrestha[1,2], Pallavi Singh[3], Jeevan B. Sherchand[4,5]

1 Center of Research for Environment, Energy and Water, Kathmandu, Nepal, 2 Interdisciplinary Center for River Basin Environment, University of Yamanashi, Takeda, Kofu, Yamanashi, Japan, 3 Nepal Red Cross Society, Head Office, Kathmandu, Nepal, 4 Tribhuvan University, Institute of Medicine, Research Directorate, Kathmandu, Nepal, 5 Nepal Health Research Council, Government of Nepal, Ethical Review Board, Kathmandu, Nepal

* rabin@creew.org.np

**Data Availability Statement:** The data are available as Supporting Information files of this paper.

**Funding:** The authors received no specific funding for this work.

## Abstract

The COVID-19 pandemic has become one of the global health challenges in the current context. In Nepal, the first confirmed case was reported on 23 January 2020, and since then, it has resulted in several negative impacts, including economic disruption and deterioration of physical and mental health. In such a pandemic, it is indispensable to understand the knowledge and behavioral patterns of the general population regarding COVID-19. Therefore, our study aimed to assess the knowledge, attitudes and practices on COVID-19 among the general population in most affected districts and its relationship with socio-demographic conditions. The cross-sectional study was conducted among the general population above the age of 18 years from eight districts of Nepal including Kathmandu, Bhaktapur, Lalitpur, Morang, Sunsari, Rupandehi, Chitwan, and Kaski. A convenient non-probability sampling method was considered with total sample size of 702. The questionnaire survey was conducted to collect data. Descriptive statistics, non-parametric statistical tests, and a logistic regression model were used for analysis. The study showed that 93.3% of respondents had knowledge of overall preventive practices, whereas only 32% had knowledge of overall symptoms of COVID-19. Regarding attitudes, only 14.3% strongly believed that the COVID-19 pandemic would end soon. The preventive practice was reduced after the lockdown compared to that of during the lockdown. The respondents with white-collar occupations, high-income, and unmarried were good at KAP. Similarly, highly educated and those residing in urban areas had good knowledge and practice. The study findings will help in the development of targeted programs to improve the general population's knowledge, attitudes and practices on COVID-19, which is paramount to dealing with the existing pandemic and also such possible future waves of the pandemic.

**Competing interests:** The authors have declared that no competing interests exist.

## Introduction

Coronavirus disease 2019 (COVID-19) is an infectious disease first identified in Wuhan, China's Hubei Province in December 2019 [1]. The World Health Organization (WHO) declared this disease a Public Health Emergency of International concern on 30 January, 2020, and later declared it a pandemic on 11 March, 2020 [2]. Now, it has been reported in almost all countries worldwide. Nepal first confirmed the case of COVID-19 on 23 January 2020 [3]. As of 12 April 2023, globally total number of cases were 762,791,152, and the total death were 6,897,025 [4]. In Nepal as of 18 April 2023 the total number of tests conducted were 6,028,151, COVID-19 cases were 1,002,346 and total death were 12,025 [5]. Considering the recent figures, as of 9 April 2023, the total number of cases were 3,000,717, and deaths were 21,149 globally in the last 28 days [6]. The statistics underscore the existence of an underlying risk of the pandemic.

The lengthening of the pandemic might result in unprecedented challenges such as economic disruption, loss of jobs and business, and increased in domestic violence among the general population [7–9]. This would ultimately affect physical health, such as discomfort, pain, lack of physical activities, and a sedentary lifestyle leading to weight gain and an increase in the risk of cardiovascular disease, mental health, such as depression as well as spiritual health [10–13]. A large number of mutations and the emergence of several variants of the concern (VOCs) of severe acute respiratory syndrome coronavirus-2 (SARS CoV-2) is highly probable in the future as well with increased transmissibility, damaging effect on human health, and increased risk of hospitalization [14]. In such circumstances, Nepal's health care system is fragile to handle an extensive increase in morbidity rates during the pandemic. Also, delivering quality health care service is very tough and challenging compared to other developed countries due to the limited resources such as skilled human resources, infrastructures, and equipment. Although, the total health budget was increased by Nepal Government during the pandemic in fiscal years 2020/21, 2021/22, the decrease in budget allocation to hospitals and academics compared to previous fiscal years has aggravated the situation [15–17]. In addition, the occurrence of high comorbidities further intensifies the complications resulting in high mortality [18–20].

Prevention and control of the spread of COVID-19 is still a challenge in the current context of emerging new variants of SARS-CoV-2. The rapidly changing information on evolving disease like COVID-19 is one of the major challenges. Though vaccines have been developed which can reduce the transmission and risk of severity of the disease [21] but cannot stop the transmission completely. Besides, it is difficult to develop herd immunity for COVID-19 despite vaccination due to the continuous emergence of new variants [22]. Moreover, the spread of disease is highly dependent on the habit of practicing preventive measures such as personal hygiene, social distancing, use of masks, etc. Hence, following these preventive measures properly along with vaccination is more effective in reducing the vulnerability of the population as well as the morbidity and mortality rates in such a challenging situation for building a disaster resilient society [23–26].

The Nepal government, in collaboration with other stakeholders, are conducting various activities such as the preparation of Information, Education, Communication (IEC) material and dissemination of information to spread knowledge about COVID-19, through various media such as social media, TV, radio, and mobile phone but the dissemination of knowledge to the community are still not satisfactory. Many marginalized and poor communities do not have access to these sources [27–30]. In addition, the people having access to these media sources are also unable to choose the authentic and correct information due to the lack of digital literacy as much of the misleading information on COVID-19 is also circulated through these sources [31]. The information-seeking behavior of people determines their risk

perception during the crisis period that further varies according to socio-demographic condition of people. On the other hand, after lifting the lockdown in the country, people's risk perception might vary due to the change in circumstances that might impact the behavior of people with increase carelessness towards the preventive practice. Generally, the low perceived risk diminishes the positive health behavior that increases the risk of disease transmission. The proper information dissemination among the people is essential for shaping risk perception of people [32,33]. However, the absence of proper coordination, well-planned, and clearly defined roles and responsibilities of 3 tiers of government (local, provincial, and federal) regarding the COVID-19 pandemic have affected the timely and efficient implementation of programs for shaping the risk perception and motivating the people for adhering preventive measures consistently [34]. Lack of timely information to improve knowledge and preventive behavior on dealing with the crisis during a pandemic can further create anxiety and fear of being discredited and treated unfairly. Consequently, the general public might be reluctant to get tested, which further delays the treatment with a large number of populations remaining undiagnosed [35,36]. Therefore, early identification of symptoms and timely treatment are vital in this scenario to control the spread of disease and fast recovery.

In such situations, it is imperative to understand what is known, what is believed and accepted, and what is done regarding the preventive measures concerning COVID-19. The necessity of a high level of psychological readiness, i.e. belief and acceptance towards the severity of disease, particularly for behavioral change [37] increased the importance of the study. In addition, previous studies conducted on Ethiopia [38–40], Bangladesh [41,42], Turkey [43], China [44–46], Malaysia [47], Iran [48], Indonesia [49], and Cameroon [50], Uganda [51], Pakistan [52], Bhutan [53] regarding knowledge, attitudes and practices (KAP) on COVID-19 also provided substantiation of clear understanding of these concerns further underscored the significance of the study. Some studies were conducted about KAP on COVID-19 in the context of Nepal as well [54–60]. However, the uniqueness of our study is it was focused on most affected districts of Nepal with a higher number of cases. The situational understanding is crucial in these areas to reduce the existing and future risk of disease outbreaks, including the rapid transmission in the study areas as well as entire country. Besides, we have included more detailed information regarding the KAP on COVID-19 compared to the previous studies. In our study, we hypothesized the variation in knowledge, attitudes and practices with the difference in socio-demographic characteristics and also variation in preventive practice during and after lockdown. Therefore, our paper was structured to discuss the situation of knowledge, attitudes and practices on COVID-19 among the general population of most affected districts and its relationship with socio-demographic characteristics. The study results will help to comprehend the knowledge gap and behavioral patterns among the general population of affected areas that are important to refine and prioritize the policy as well as allocate the resources for the implementation of intervention programs to address the existing challenges of the COVID-19 pandemic. Besides, the information collected during such a crisis is imperative to improve the competency of authorities and policymakers to respond effectively not only during current pandemics but also aid in handling any potential risk of an outbreak in the future to reduce the cascading impacts.

## Methods

### Study area and sampling

The cross-sectional study was conducted among the general population above the age of 18 years from eight districts- Kathmandu, Bhaktapur, Lalitpur, Morang, Sunsari, Rupandehi, Chitwan, and Kaski of Nepal (Fig 1).

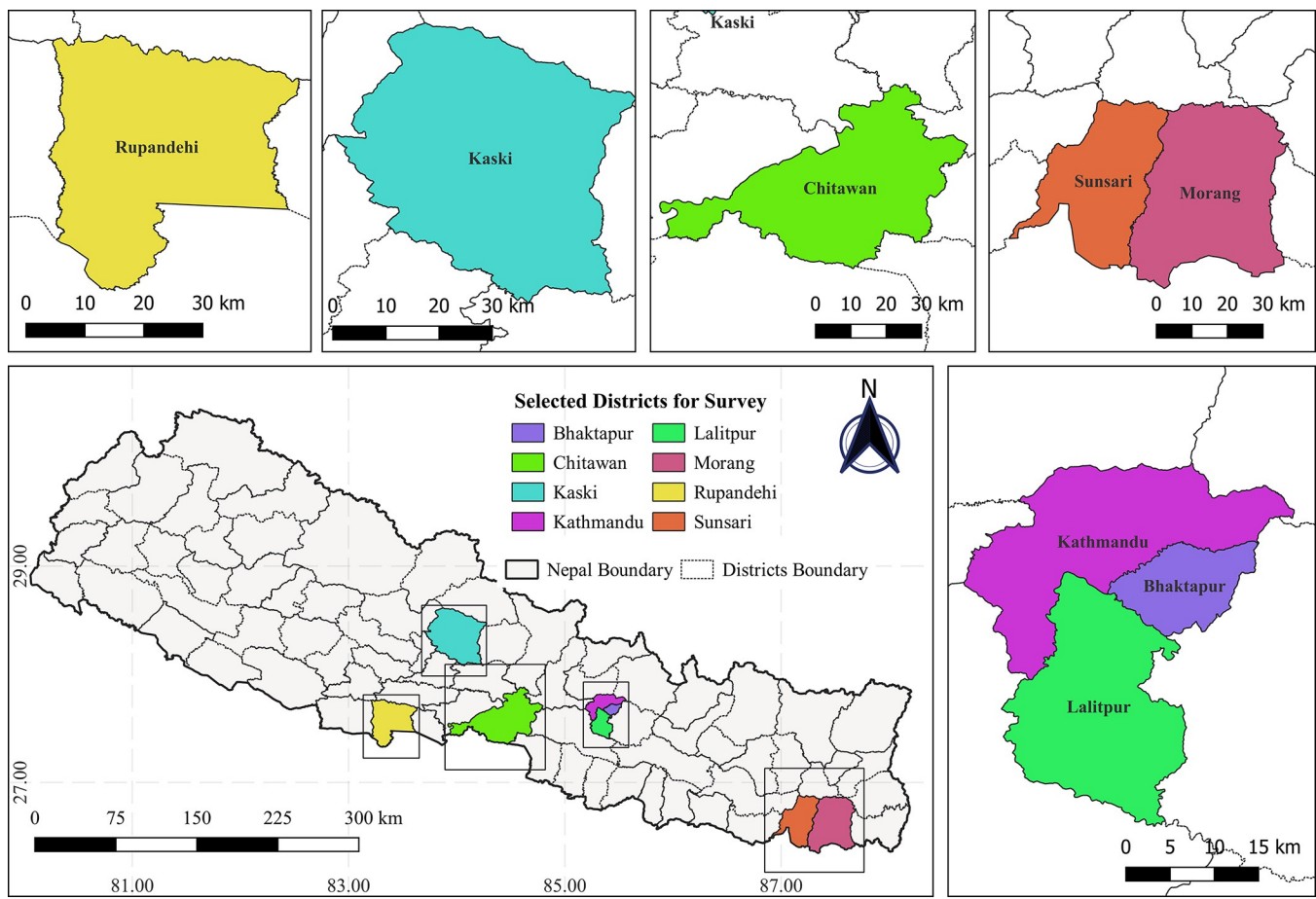

**Fig 1. Location of the selected districts (Bhaktapur, Chitwan, Kaski, Kathmandu, Lalitpur, Morang, Rupandehi, and Sunsari).** The darkest grey thick lines covering the grey area represent the country boundary of Nepal and the soft grey line inside represent the administrative boundaries of Nepal separating different districts. The Location of different districts is presented by rectangles in Nepal. Country boundary and Administrative boundaries shape file of Nepal, retrieved from Administrative Boundaries Reference (view layer) of FAO (https://data-in-emergencies.fao.org/datasets/hqfao::administrative-boundaries-reference-view-layer/explore?layer=0), under an open license (CC BY 4.0): https://data-in-emergencies.fao.org/datasets/hqfao::administrative-boundaries-reference-view-layer/about?layer=0. Figure created with QGIS software version 3.28, an open source Geographic Information System (GIS) licensed under the GNU General Public License (https://bit.ly/2BSPB2F).

The sample size (n) was determined by following the formula given by Arkin and Colton (1963) [61].

$$n = \frac{N * z^2 * P(1 - P)}{N * d^2 + z^2 * P(1 - P)}$$

Where,

n = sample size

N = Total population size (total population of eight selected districts) = 61, 98,158 [62].

Z = confidence interval (at 95% level z = 1.96)

P = estimated population proportion (0.5)

d = error limit of 5% (0.05)

The minimum sample size required for the study was 400.

A convenient non-probability sampling method was considered due to the inappropriateness of the probability sampling during the period of the pandemic. The total sample collected was 702.

The number of samples in each of the eight selected districts was determined according to the proportion of the population of the districts. The sample size for each district ($n_d$) was calculated as follows:

Proportion of population in each district (A) = (Population of a district)/ (Total population of all eight districts) ×100%.

Sample size of each district ($n_d$) = (A)/100×Total sample size (702).

Hence, the number of participants selected in each district were Kathmandu-197, Bhaktapur-35, Lalitpur-5, Morang-107, Sunsari-88, Rupandehi-99, Chitwan-66, and Kaski-56.

## Data collection

For data collection, we used a mixed method approach which included phone survey and face to face interview as it was difficult to interact and collect all the samples employing only one approach during the pandemics. Skilled volunteers associated with Nepal Red Cross Society (NRCS) district chapters were mobilized to conduct surveys. The data was collected from 21 March 2021 to 12 April 2021.

## Ethical concern

The study was approved by the Ethical Review Board of the Nepal Health Research Council with ethical approval no.-Ref no. 2240. The verbal and written Informed consent were taken from all the respondents who participated in the study through telephone survey and face to face interview respectively.

## Measurements

The questionnaire has four domains- knowledge, attitudes, practices and socio-demographic characteristics.

## Knowledge, attitudes, and practices (KAP)

The questionnaire contained different items for determining 'knowledge' level, 'attitude' level, and 'practice' level of the respondents. The respondents were tested on their 'knowledge' level for twelve different issues. Questions related to knowledge of 'symptoms', 'preventive measures', 'pre-condition to stay in quarantine', 'diet and exercise', 'transmission of SARS-CoV-2 through aerosols', 'survival duration of SARS-CoV-2 in fomites (plastic and stainless steel, cardboard, copper)' and aerosols', 'awareness on transmission through mobile phone', 'new variants of SARS-CoV-2' and 'co-morbidities of several chronic diseases' were included in the questionnaire. Thirteen binary questions (yes/ no) questions were used to assess the knowledge of 'symptoms' with a total score range of 0–13 which was dichotomized with a cutoff value of 10 (median) as those having good knowledge ($\geq$10) and poor knowledge ($<$10). There were six binary questions related to knowledge on 'preventive measures' with a total score in the range 0–6. The respondents with knowledge on all the preventive measures were categorized as having good knowledge and others with poor knowledge. Similarly, there were five binary questions related to knowledge on 'precondition to stay in quarantine'. The respondents with knowledge on all pre-condition to stay in quarantine were categorized as having good knowledge and others as having poor knowledge. The knowledge level on co-morbidities included eight diseases. The score ranged from 0–8 and was dichotomized with cutoff value of

6 (median) as having good knowledge (≥6) and poor knowledge (<6). The questions regarding the knowledge on 'diet and exercise', transmission of SARS-CoV-2 through aerosols, life duration of SARS-CoV-2 in fomites (plastic and stainless steel, cardboard, copper) and aerosols, 'awareness on transmission through mobile phone', 'new variants of SARS-CoV-2' were simply responded as 'yes' or 'no'. Lastly, the scores on these twelve issues were summed to measure the overall knowledge level of the general population with total score ranging from 0–12. The knowledge level was dichotomized to overall good (≥7) and poor knowledge (<7) on COVID-19 with a cutoff value of 7 (median). The attitude questionnaire included 'a willingness to adhere preventive measures', 'perception of the importance of staying in quarantine', 'self-safety against disease' and 'proper diet and exercise', 'concerned about the careful steps to reduce the difficult situation' and 'belief in end of pandemic shortly' Attitude level was measured using a five-point Likert Scale type of questions where the responses varied from strongly (1) disagree to strongly agree (5). The total score of the attitude questionnaire was the sum of scores of all the items ranging from 6 to 30, which was categorized into good (≥26) and a poor attitude (<26) with a cutoff value of 26 (median).

The practice questionnaire included 'adherence to hand washing or use of alcohol-based sanitizer', 'covering mouth and nose while coughing and sneezing', 'social distancing', 'use of mask while going outside', 'avoid crowd', 'stay home', 'cleaning mobile phone with sanitizer' and 'having proper diet and exercise'. Practice level after the lockdown was measured using a five-point Likert Scale questionnaire for prevention of COVID-19 from never (1) to always (5). The total score of practice questionnaires ranging from 9 to 45 was categorized as good practice (≥33) and poor practice (<33), with a cutoff value of 33 (median).

## Socio-demographic characteristics

The structured questionnaire was used to collect the data on socio-demographic characteristics such as geographical location (rural municipality, urban municipality), gender (male, female), age (<20 years, 20–30 years, 31–40 years, 41–50 years, >50 years), marital status (married, unmarried), education (no education, literate, basic education, secondary education, undergraduate, graduate and above), income (<NPR 5000, NPR 5000–10,000, NPR 10,000–15,000, NPR 15,000–20,000, NPR >20,000), occupation of the respondent (white-collar occupation—service, business, house rent; blue-collar occupation-agriculture, labor; and others—self-employed, remittance and others).

The questionnaire was developed and modified from previous studies conducted in Bangladesh [41] and China [46], and several other relevant literatures were considered [63–70]. The questionnaire was first developed in the English language, translated to Nepali language and back translated to English language. The translation and back translation was done by experts. The questionnaire was then pre-tested among 30 participants, and the feedbacks were incorporated to develop the final questionnaire.

## Statistical analysis

The results on the responses of 'knowledge', 'attitude', and 'practice' level were described using descriptive statistics. Chi-square test was used to evaluate the differences in knowledge, attitude and practice in relation to socio-demographic characteristics. The data did not follow normal distribution (Shapiro-Wilk test, p < 0.05) showing non-normal distribution of continuous variable. Therefore, the Wilcoxon Signed Rank Test was used to assess the differences in 'practice for prevention before and after lockdown' among the respondents. The reliability coefficient (Cronbach's Alpha) for the questionnaire used to measure knowledge, attitudes, and practices were 0.80, 0.602 and 0.82 respectively which is was considered as acceptable and

reliable [71]. A multivariate logistic regression model was used with a 95% confidence interval to find the relationship of KAP with socio-demographic characteristics. The significance level was set at p < 0.05 for all the statistical analysis. Statistical analyses were conducted using Statistical Package for the Social Sciences V.20 (SPSS Inc., Chicago, Illinois, USA).

## Results

### Knowledge

The study revealed that very few proportions of respondents had knowledge of the symptoms of conjunctivitis (n = 159, 22.9%) followed by skin rashes or discoloration of skin or toes (n = 172, 24.7%), diarrhea (n = 258, 37%), whereas, a higher proportion of respondents had knowledge on fever (n = 699, 99.7%), dry cough (n = 680, 97%) and tiredness (n = 634, 90.3%), (Fig 2).

Most of the respondents had good knowledge regarding several preventive measures. Higher proportion of them had good knowledge on covering of nose and mouth while sneezing and coughing (n = 700, 99.7%), using mask (n = 698, 99.4%) and necessity of washing hands or use of alcohol-based sanitizer (n = 692, 98.6%), (Fig 3).

Similarly, a higher proportion of respondents had knowledge of pre-conditions to stay in quarantine if come into contact with infected person, such as contact with droplets while sneezing and coughing (n = 675 96.4%) followed by taking care of an infected person within the house (n = 649, 93%) and using utensils of an infected person (n = 630, 90.4%) (Fig 4).

Considering the knowledge of increase in risk of death due to comorbidities among COVID-19 patients, many of the respondents had knowledge of increased risk due to lung disease (n = 604, 86.8%), followed by cardiovascular symptoms (n = 511, 73%). At the same time, very few had knowledge of increased risk due to cerebrovascular disease (n = 292, 41.8%) (Fig 5).

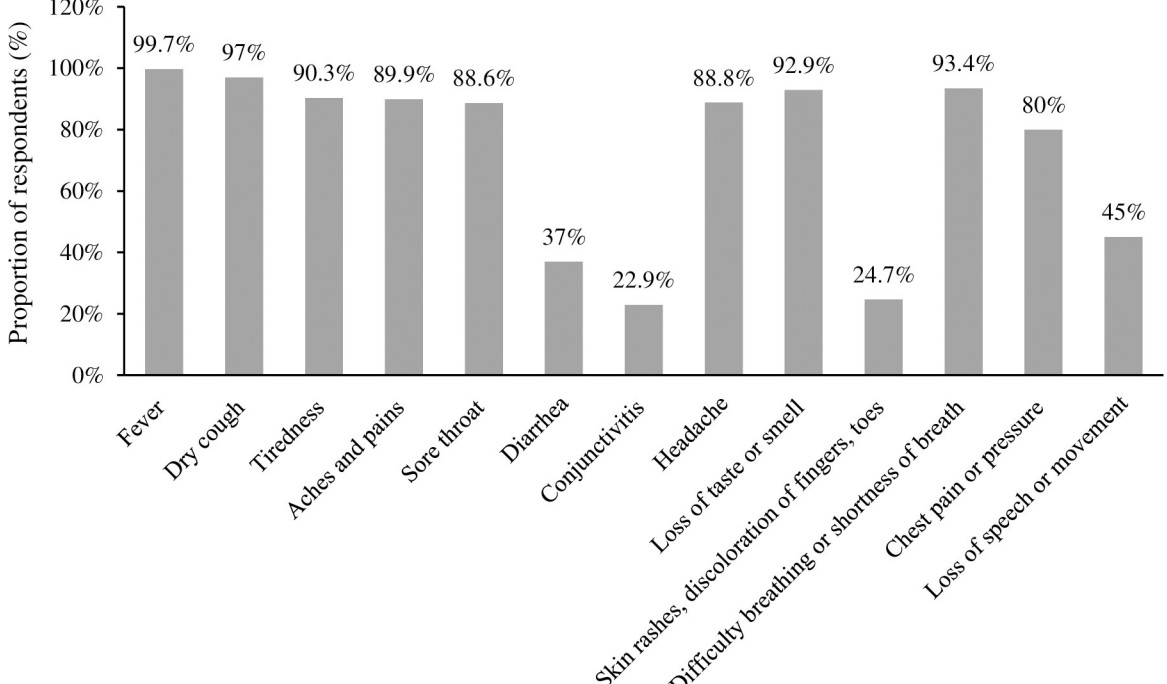

**Fig 2. Knowledge of symptoms of the COVID-19 pandemic among the respondents.**

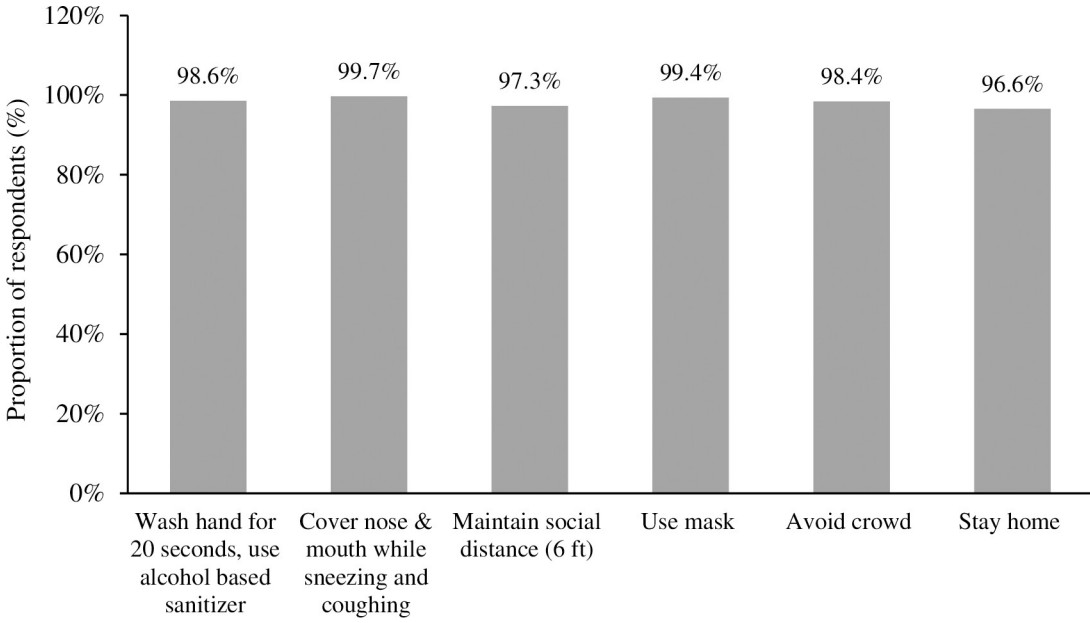

**Fig 3. Knowledge of prevention on COVID-19 among the respondents.**

Table 1 comprising the overall knowledge of COVID-19 among the respondents, further revealed that a significant proportion of them (n = 655, 93.3%) had good knowledge about overall preventive measures regarding COVID-19. However, very few respondents had good knowledge of symptoms (n = 221, 32%) and risk of co-morbidities among the COVID-19 infected patient (n = 283, 40.7%). Similarly, (n = 466, 67.1%) of respondents had overall

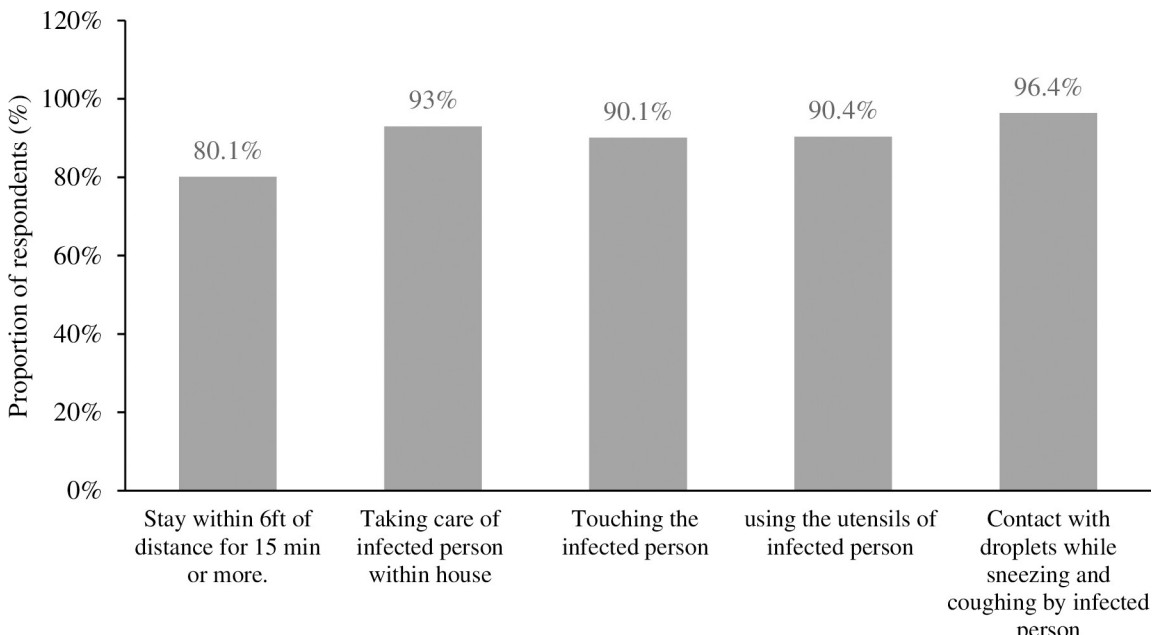

**Fig 4. Knowledge of pre-condition to stay in quarantine among the respondents if come into contact with infected person.**

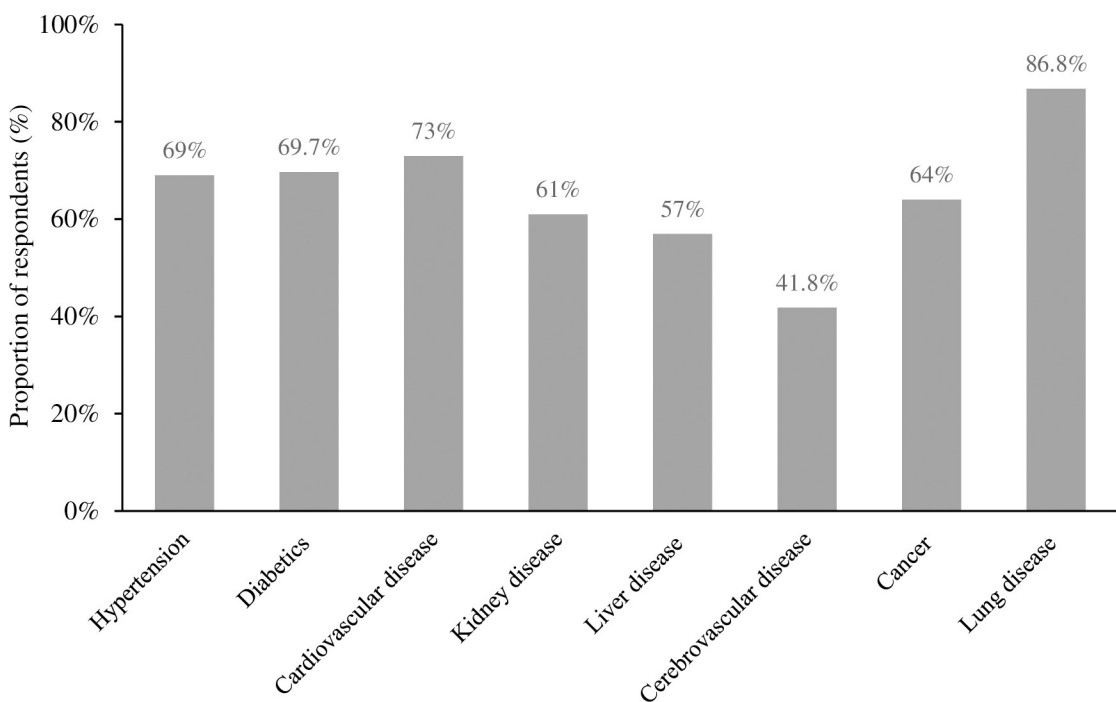

**Fig 5. Knowledge of respondents regarding increase in risk of death due to co-morbidities among the COVID-19 infected person.**

knowledge of the pre-condition of quarantine if come into contact with COVID-19 infected person. Concerning the knowledge of survival of SARS CoV-2 in fomites, (n = 375, 53.6%) knew about its survival duration on plastic and stainless steel, (n = 280, 40%) on cardboard, (n = 306, 43.7%) on copper, and (n = 352, 50.2%) on aerosols.

**Table 1. Knowledge level of respondents on COVID-19.**

| S.N. | Items | Proportion (%) |
|---|---|---|
| K1 | Good Knowledge of the symptoms of COVID-19 as mentioned by WHO | 221 (32%) |
| K2 | Good knowledge of preventive measures to be protected from COVID-19 | 655 (93.3%) |
| K3 | Good knowledge of pre-conditions to stay in quarantine if come into contact with COVID-19 infected person | 466 (67.1%) |
| K4 | Familiarity with significance of proper diet and exercise that will help to boost the immune system to fight against COVID-19 | 653 (93.4%) |
| K5 | Familiarity with transmission of SARS-CoV-2 from aerosols | 429 (61.5%) |
| K6 | Knowledge of seventy-two hours survival duration of SARS-CoV-2 in plastic, and stainless steel | 375 (53.6%) |
| K7 | Knowledge of twenty-four hours survival duration of SARS-CoV-2 in cardboard | 280 (40%) |
| K8 | Knowledge of four hours survival duration of SARS-CoV-2 in copper | 306 (43.7%) |
| K9 | Knowledge of three hours survival duration of SARS-CoV-2 in aerosols | 352 (50.2%) |
| K10 | Awareness regarding transmission of SARS-CoV-2 virus through surface of mobile phones that mostly comes into contact with the nose and mouth increase the risk of infection | 355 (51.2%) |
| K11 | Knowledge of new variants of SARS-CoV-2 | 421 (60.5%) |
| K12 | Good knowledge on co-morbidities of several chronic diseases among the COVID-19 infected persons increase the death rate | 283 (40.7%) |

**Table 2. Attitudes of respondents on COVID-19.**

| S.N. | Items | Strongly agree | Agree | Neutral | Disagree | Strongly disagree |
|---|---|---|---|---|---|---|
| A1 | I am willing to adhere the all the preventive measures to reduce the risk of transmission. | 293 (41.7%) | 320 (45.6%) | 82 (11.7%) | 3 (0.4%) | 4 (0.6%) |
| A2 | I believe staying in quarantine if come in contact with infected person will reduce the transmission. | 338 (48.1%) | 296 (42.2%) | 63 (9%) | 4 (0.6%) | 1 (0.1%) |
| A3 | I think keeping myself safe is crucial in breaking the chain of transmission of COVID-19. | 473 (67.4%) | 199 (28.8%) | 24 (3.4%) | 4 (0.6%) | 2 (0.3%) |
| A4 | I believe that exercise and diet will increase immunity to fight against the COVID-19. | 424 (61.2%) | 214 (30.9%) | 41 (5.9%) | 5 (0.7%) | 9 (1.3%) |
| A5 | I am anxious that it will lead to a difficult situation with further loss of life and property if failed to take careful steps on time. | 272 (39.1%) | 353 (50.7%) | 59 (8.5%) | 9 (1.3%) | 3 (0.4%) |
| A6 | I believe the pandemic will end soon. | 99 (14.3%) | 215 (31.1%) | 282 (40.8%) | 56 (8.1%) | 39 (5.6%) |

## Attitudes

Regarding attitudes, a significant proportion of respondents (n = 473, 67.4%) strongly agreed that keeping themselves safe was crucial for breaking the chain of transmission of COVID-19 and (n = 424, 61.2%) strongly agreed that exercise and diet would increase the immunity to fight against COVID-19. However, a very small proportion of respondents (n = 99, 14.3%) strongly agreed that the pandemic would end soon (Table 2).

## Practices

During the lockdown, a higher proportion of respondents (n = 601, 86.1%) always used mask while going outside, followed by those who always covered their mouth and nose while coughing and sneezing (n = 576, 82.5%) and always washed their hands with soap for 20 sec or used alcohol-based sanitizer (n = 522, 74.6%). Nevertheless, very few respondents did regular exercise (n = 222, 31.7%) and cleaned mobile phones with sanitizer after coming from outside (n = 225, 32.2%).

After lockdown, (n = 439, 63.6%) always covered the mouth and nose while coughing and sneezing. However, very few of them always stayed at home for safety (n = 53, 7.6%), cleaned mobile phones with sanitizer after coming from outside (n = 57, 8.2%), and avoided crowds (n = 58, 8.4%) (Table 3). The median score for practice for prevention during lockdown was 40 and after the lockdown was 33. The Wilcoxon Rank Test showed a statistically significant difference in practice for prevention during and after lockdown (p<0.001).

## Relationship of KAP and socio-demographic characteristics

The good knowledge on COVID-19 was significantly higher among the residents of urban areas (n = 362, 60.9%) (p<0.001), unmarried (n = 145, 67.8%) (p<0.01) following white-collar occupation (n = 255, 6.4%) (p<0.001), educational level with graduate and above (n = 57, 87.7%) (p<0.001) and high-income level (n = 312, 72.2%) (p<0.001).

Good attitudes were significantly higher among the unmarried respondents (n = 134, 63.5%) (p<0.001), those following the white-collar occupation (n = 233, 57.4%) (p<0.001), and high-income group (n = 265, 59.7%) (p<0.001). In the same way, good practices for the prevention of COVID -19 were significantly higher among the respondents following white-collar occupation (n = 243, 60%) (p<0.001), unmarried group (n = 122, 57.5%) (p<0.05), high-educated group (n = 46, 66.7%) (p<0.001) and having high-income (n = 279, 63.1%) (p<0.001). The proportion of respondents with good practices was higher (n = 324, 52.7%) among urban residents though not statistically significant (Table 4).

**Table 3. Practices of prevention among respondents during and after the lockdown.**

| S.N. | Items | Status | Always | Most of the time | Sometime | Rarely | Never |
|------|-------|--------|--------|------------------|----------|--------|-------|
| P1 | Washing hands with soap for 20 sec or use alcohol-based sanitizer | DL | 522 (74.6%) | 139 (19.9%) | 26 (3.7%) | 8 (1.1%) | 5 (0.7%) |
|    |                                                                  | AL | 294 (42.5%) | 239 (34.6%) | 105 (15.2%) | 35 (5.1) | 18 (2.6%) |
| P2 | Cover mouth and nose while coughing and sneezing. | DL | 576 (82.5%) | 98 (14%) | 15 (2.1%) | 5 (0.7%) | 4 (0.6%) |
|    |                                                   | AL | 439 (63.6%) | 169 (24.5%) | 58 (8.4%) | 13 (1.9%) | 11(1.6%) |
| P3 | Maintain social distance up to 6ft | DL | 469 (67%) | 177 (25.3%) | 37 (5.3%) | 12 (1.7%) | 5 (0.7%) |
|    |                                    | AL | 95 (13.7%) | 226 (32.7%) | 241 (34.9%) | 89 (12.9%) | 40 (5.8%) |
| P4 | Use a mask while going outside | DL | 601 (86.1%) | 72 (10.3%) | 15 (2.1%) | 6 (0.9%) | 4 (0.6%) |
|    |                                | AL | 419 (60.5%) | 168 (24.3%) | 61 (8.8%) | 28 (4%) | 16 (2.3%) |
| P5 | Avoid crowd | DL | 466 (66.7%) | 183 (26.2%) | 25 (3.6%) | 19 (2.7%) | 6 (0.9%) |
|    |             | AL | 58 (8.4%) | 250 (36.3%) | 264 (38.4%) | 97 (14.1%) | 19 (2.8%) |
| P6 | Stay at home for safety | DL | 502 (71.7%) | 149 (21.3%) | 20 (2.9%) | 22 (3.1%) | 7 (1%) |
|    |                         | AL | 53 (7.6%) | 216 (31.2%) | 247 (35.6%) | 139 (20.1%) | 38 (5.5%) |
| P7 | Clean mobile phone with sanitizer after coming from outside | DL | 225 (32.2%) | 108 (15.5%) | 103 (14.7%) | 69 (9.9%) | 194 (27.8%) |
|    |                                                             | AL | 57 (8.2%) | 70 (10.1%) | 150 (21.7%) | 112 (16.2%) | 302 (43.7%) |
| P8 | Having proper diet | DL | 433 (61.9%) | 199 (28.5%) | 32 (4.6%) | 24 (3.4%) | 11 (1.6%) |
|    |                    | AL | 321(46.3%) | 232 (46.3%) | 92 (13.3%) | 32 (4.6%) | 16 (2.3%) |
| P9 | Doing regular exercise | DL | 222 (31.7%) | 160 (22.9%) | 164 (23.4%) | 31 (4.4%) | 123 (17.6%) |
|    |                        | AL | 138 (19.9%) | 135 (19.5%) | 194 (28%) | 69 (10%) | 157 (22.7%) |

DL-During lockdown; AL-After lockdown.

Logistic regression analysis showed that highly educated respondents such as graduate and above (OR: 17.08, 95% CI 5.19 to 56.12), undergraduate (OR: 8.88, 95% CI 3.41 to 23.10), having secondary education (OR: 4.06, 95% CI 1.66 to 9.95) had higher odds of having knowledge than the non-educated group. Likewise, the odds ratio of having good knowledge of prevention among the higher income group with monthly income >NRs 20,000 was 3.52 (1.24 to 10.39).

Regarding attitudes, people with white-collar occupations 1.64 (1 to 2.70) and other occupations 1.57 (0.89 to 2.77) had high odds of having good attitudes compared to those having a blue-collar occupation.

In the same way, females were 1.63 times more likely to have good preventive practices (95% CI 1.14 to 2.35) than males. Our study further revealed that the odds ratio of practicing good preventive behavior was 1.83 times (95% CI 1.07 to 3.12) for those with white-collar occupations. Moreover, educated respondents such as graduate and above (OR: 8.79, 95% CI 2.97 to 25.97), undergraduate (OR: 8.36, 95% CI 3.14 to 22.26), secondary education (OR: 8.15, 95% CI 3.17 to 20.93) and basic education (OR: 4.08, 95% CI 1.15 to 10.98) had higher odds of practicing good preventive behavior compared to non-educated group (Table 5).

## Discussion

Our study aimed to understand the knowledge, attitudes, and practices (KAP) on COVID-19 and its relationship with socio-demographic characteristics among the general population in most affected districts of Nepal.

The higher proportion of respondents had good knowledge of symptoms such as fever, dry cough, tiredness, aches and pains, sore throat, loss of taste or smell, difficulty breathing or shortness of breath, and very few had knowledge of symptoms such as diarrhea, conjunctivitis, rash on the skin, or discoloration of fingers or toes and loss of speech or movement. The knowledge of the overall symptoms of COVID-19 among the community was observed to be very low, although understanding is essential during the pandemic. Other similar studies also

**Table 4. Socio-demographic characteristics of the respondents with different levels of KAP on COVID-19.**

| Variables | Knowledge | | Chi-square test (p-value) | Attitudes | | Chi-square test (p-value) | Practices | | Chi-square test (p-value) |
|---|---|---|---|---|---|---|---|---|---|
| | Number of respondents | Good Knowledge (%) | | Number of respondents | Good Attitude (%) | | Number of respondents | Good Practice (%) | |
| **Municipality** | | | | | | | | | |
| Rural | 62 | 21 (33.9%) | | 62 | 27 (43.5%) | 0.11 | 60 | 24 (40%) | .06 |
| Urban | 594 | 362 (60.9%) | <0.001 | 615 | 332 (54%) | | 615 | 324 (52.7%) | |
| **Gender** | | | | | | | | | |
| Male | 312 | 189 (60.5%) | 0.3 | 318 | 152 (47.8%) | 0.01 | 326 | 163 (50%) | 0.35 |
| Female | 350 | 199 (56.9%) | | 364 | 208 (57.1%) | | 355 | 190 (53.5%) | |
| **Marital status** | | | | | | | | | |
| Unmarried | 214 | 145 (67.8%) | 0.001 | 211 | 134 (63.5%) | <0.001 | 212 | 122 (57.5%) | 0.04 |
| Married | 446 | 241 (54%) | | 469 | 225 (48%) | | 467 | 230 (49.3%) | |
| **Main occupation** | | | | | | | | | |
| Blue-collar | 115 | 42 (36.5%) | <0.001 | 120 | 43 (35.8%) | <0.001 | 118 | 32 (27.1%) | <0.001 |
| White-collar | 396 | 255 (6.4%) | | 406 | 233 (57.4%) | | 405 | 243 (60%) | |
| Others | 118 | 73 (61.9%) | | 123 | 68 (55.3%) | | 123 | 61 (49.6%) | |
| **Age (years)** | | | | | | | | | |
| <20 | 32 | 20 (62.5%) | 0.03 | 29 | 17 (58.6%) | 0.007 | 33 | 16 (48.5%) | 0.12 |
| 20–30 | 256 | 162 (63.3%) | | 256 | 153 (59.8%) | | 249 | 134 (53.8%) | |
| 31–40 | 153 | 85 (55.5%) | | 156 | 84 (53.8%) | | 157 | 73 (46.5%) | |
| 41–50 | 120 | 74 (61.7%) | | 132 | 63 (47.7%) | | 134 | 80 (59.7%) | |
| >50 | 99 | 45 (45.5%) | | 106 | 42 (39.6%) | | 105 | 48 (45.7%) | |
| **Education** | | | | | | | | | |
| No Education | 49 | 10 (20.4%) | | 51 | 21 (41.2%) | 0.02 | 54 | 8 (14.8%) | <0.001 |
| Literate | 19 | 7 (36.8%) | <0.001 | 23 | 8 (34.8%) | | 22 | 9 (40.9%) | |
| Basic Education | 83 | 26 (31.3%) | | 87 | 33 (37.9%) | | 87 | 29 (33.3%) | |
| Secondary Education | 220 | 116 (52.7%) | | 225 | 127 (56.4%) | | 221 | 118 (53.4%) | |
| Undergraduate | 224 | 170 (75.9%) | | 225 | 134 (59.6%) | | 225 | 140 (62.22%) | |
| Graduate and above | 65 | 57 (87.7%) | | 68 | 35 (51.5%) | | 69 | 46 (66.7% | |
| **Income-Nepalese Rupee (NPR)** | | | | | | | | | |
| <5000 | 25 | 7 (28%) | <0.001 | 28 | 7 (25%) | <0.001 | 27 | 7 (25.9%) | <0.001 |
| 5000–10,000 | 48 | 9 (18.7%) | | 47 | 16 (34%) | | 48 | 11 (22.9%) | |
| 10,000–15,000 | 54 | 16 (29.6%) | | 60 | 23 (38.3%) | | 59 | 12 (20.3%) | |
| 15,000–20,000 | 100 | 43 (43%) | | 100 | 48 (48%) | | 102 | 44 (43.1%) | |
| >20,000 | 432 | 312 (72.2%) | | 444 | 265 (59.7%) | | 442 | 279 (63.1%) | |

revealed that a higher proportion of community people had knowledge of symptoms such as fever and cough and very less knew about the other symptoms of COVID-19 compared to our study [38,41,55,72]. A lower level of knowledge on overall symptoms might prevent the early detection of disease and hinder the timely treatment [36].

Our study illustrated that knowledge of preventive behaviors seemed to be high, which is in line with a similar study conducted in Bangladesh [41], China [45], and Nepal [54]. However, the knowledge of the survival period of SARS-CoV-2 in an external environment was low, which might increase the risk of disease transmission. Similarly, the low knowledge associated with the risk of comorbidities such as diabetes, Chronic Obstructive Pulmonary Disease, hypertension, malignancy, etc. among the study population can further augment the severity

**Table 5. Odds ratios of having good knowledge, attitudes and practices on COVID-19 among the respondents according to the socio-demographic characteristics.**

| Variables | Good knowledge | | Good attitudes | | Good practices | |
|---|---|---|---|---|---|---|
| | Univariate | Multivariate | Univariate | Multivariate | Univariate | Multivariate |
| **Municipality** | | | | | | |
| Rural | 1 | 1 | 1 | 1 | 1 | 1 |
| Urban | 3.04 (1.76 to 5.28)*** | 2.29 (1.20 to 4.35)* | 1.52 (0.89 to 2.57) | 1.38 (0.77 to 2.47) | 1.67 (0.97 to 2.86) | 1.34 (0.73 to 2.46) |
| **Gender** | | | | | | |
| Male | 1 | 1 | 1 | 1 | 1 | 1 |
| Female | 0.85 (0.62 to 1.17) | 1.01 (0.69 to 1.50) | 1.45 (1.07 to 1.97)* | 1.39 (0.98 to 1.96) | 1.15 (0.85 to 1.55) | 1.63 (1.14 to 2.35)** |
| **Marital status** | | | | | | |
| Unmarried | 1 | 1 | 1 | 1 | 1 | 1 |
| Married | 0.55 (0.39 to 0.78)** | 0.90 (0.51 to 1.58) | 0.53 (0.38 to 0.74)*** | 0.59 (0.35 to 0.99)* | 0 0.71 (0.51 to 0.99)* | 0.63 (0.37 to 1.07) |
| **Main occupation** | | | | | | |
| Blue-collar | 1 | 1 | 1 | 1 | 1 | 1 |
| White-collar | 3.14 (2.04 to 4.84)*** | 1.03 (0.59 to 1.81) | 2.41 (1.58 to 3.67)*** | 1.64 (1 to 2.70)* | 4.03 (2.56 to 6.33)*** | 1.83 (1.07 to 3.12)* |
| Others | 2.82 (1.65 to 4.79)*** | 1.54 (0.82 to 2.92) | 2.21 (1.32 to 3.70)** | 1.57 (0.89 to 2.77) | 2.64 (1.54 to 4.52)*** | 1.53 (0.83 to 2.82) |
| **Age (years)** | | | | | | |
| <20 | 1 | 1 | | 1 | 1 | 1 |
| 20–30 | 1.03 (0.48 to 2.21) | 1.01 (0.39 to 2.58) | 1.04 (0.48 to 2.28) | 1.66 (0.68 to 4.04) | 1.23 (0.59 to 2.56) | 1.30 (0.55 to 3.08) |
| 31–40 | 0.75 (0.34 to 1.64) | 1.06 (0.36 to 3.12) | 0.82 (0.36 to 1.83) | 1.68 (0.61 to 4.59) | 0.92 (0.43 to 1.95) | 1.58 (0.58 to 4.30) |
| 41–50 | 0.96 (0.43 to 2.15) | 1.77 (0.57 to 5.53) | 0.64 (0.28 to 1.45) | 1.46 (0.51 to 4.17) | 1.57 (0.73 to 3.38) | 3.18 (1.11 to 9.11)* |
| >50 | 0.50 (0.22 to 1.13) | 0.50 (0.22 to 1.13) | 0.46 (0.20 to 1.06) | 1.07 (0.36 to 3.17) | 0.89 (0.40 to 1.95) | 2.44 (0.82 to 7.23) |
| **Education** | | | | | | |
| No Education | 1 | 1 | 1 | 1 | 1 | 1 |
| Literate | 2.27 (0.71 to 7.27) | 2.09 (0.56 to 7.69) | 0.76 (0.27 to 2.12) | 0.61 (0.19 to 1.95) | 3.98 (1.28 to 12.27)* | 4.18 (1.16 to 15.05)* |
| Basic Education | 1.77 (0.77 to 4.10) | 1.90 (0.73 to 4.92) | 0.87 (0.43 to 1.76) | 0.77 (0.34 to 1.69) | 2.87 (1.20 to 6.88)* | 4.08 (1.15 to 10.98)** |
| Secondary Education | 4.35 (2.06 to 9.14)*** | 4.06 (1.66 to 9.95)** | 1.85 (0.99 to 3.43) | 1.34 (0.64 to 2.80) | 6.58 (2.97 to 14.60)*** | 8.15 (3.17 to 20.93)*** |
| Undergraduate | 12.27 (5.74 to 26.32)*** | 8.88 (3.41 to 23.10)*** | 2.10 (1.13 to 3.90)* | 0.90 (0.40 to 1.99) | 9.47 (4.26 to 21.02)*** | 8.36 (3.14 to 22.26)*** |
| Graduate and above | 27.78 (10.07 to 76.67)*** | 17.08 (5.19 to 56.12)*** | 1.51 (0.72 to 3.15) | 0.56 (0.22 to 1.40) | 11.50 (4.66 to 28.35)*** | 8.79 (2.97 to 25.97)*** |
| **Income-Nepalese Rupees (NPR)** | | | | | | |
| <5000 | 1 | 1 | 1 | 1 | 1 | 1 |
| 5000–10,000 | 0.59 (0.19 to 1.84) | 0.82 (0.23 to 2.90) | 1.54 (0.54 to 4.41) | 1.80 (0.57 to 5.67) | 0.84 (0.28 to 2.53) | 0.83 (0.24 to 2.84)) |
| 10,000–15,000 | 1.08 (0.37 to 3.09) | 1.07 (0.32 to 3.55 | 1.86 (0.68 to 5.07) | 1.74 (0.58 to 5.26) | 0.72 (0.25 to 2.12) | 0.56 (0.17 to 1.87) |
| 15,000–20,000 | 1.94 (0.74 to 5.05) | 1.81 (0.60 to 5.46) | 2.76 (1.01 to 7.09) | 2.66 (0.93 to 7.58) | 2.16 (0.84 to 5.58) | 1.62 (0.54 to 4.77) |
| >20,000 | 6.68 (2.72 to 16.41)*** | 3.52 (1.24 to 10.39)* | 4.44 (1.84 to 10.66) | 3.75 (1.37 to 10.28)* | 4.89 (2.02 to 11.81)*** | 2.34 (0.83 to 6.58) |

*p < 0.05, **p < 0.01, p<0.00.

of COVID-19 patients leading to hospitalization, admission to intensive care unit, and increase in the mortality rate [70,73,74].

Our study further showed that the majority of people had a positive attitudes. Most of them agree that preventive measures will reduce the risk of transmission, need of staying in quarantine if come into contact with infected person, keeping themselves safe. The aforementioned

studies also showed similar results [53–55,36,70]. Moreover, a significant proportion of them believe in the importance of proper diet and exercise to boost the immune system, and are concerned about the need for careful and urgent steps as well as seriousness towards the pandemic to reduce further losses. The previous studies conducted in China [46], Nepal [54,55] and Pakistan [52] revealed that the majority of people agreed the pandemic could be overcome. However, our study demonstrated that a substantial number of respondents were doubtful about ending of pandemic very soon. The reason might be the fact about history of previous pandemics that revealed the health crisis lasted for years with massive loss of life and property had created the fear of prolonging the COVID-19 pandemic [75,76]. Although, many researches are carried out, and much information is explored regarding SARS-CoV-2, the virus is constantly changing and, there are still many unknown facts regarding the risk of the spread of new variants of concern that lead to uncertainties [77]. Moreover, the lack of a strong control mechanisms during health emergencies in developing countries has further diminished the people's trust upon the government and responsible authorities [78].

Preventive practices such as washing hands with soap, covering the mouth and nose while coughing and sneezing, using of mask while going outside and having proper diet and exercise were high during and after the lockdown. Several other studies have also highlighted the higher proportion of respondents adhering to hand washing with soap [41,45,47,50,54,55,57,60], covering mouth and nose while sneezing and coughing [45,55,57], and use of mask [45,47,50,53–55,57,60]. On the other hand, the study conducted in Iran stated less use of masks and covering of nose and mouth while sneezing and coughing [48]. In contrast to our study findings, doing exercise is comparatively higher among the Chinese public [45]. However, after the lockdown, the preventive practices were observed to be lowered compared to the practice during the lockdown. The reduction in the morbidity and mortality rate does not mean the pandemic will end soon. There is always the risk of the emergence of new VOCs that might be responsible for new waves of the COVID-19 pandemic. The negligence of community people was observed though they had knowledge and information. The result was in line with the study conducted in Janakpur, Nepal which revealed that though the perception of COVID-19 was not so different, the preventive behaviors were reduced in post-lockdown compared to the behaviors during the lockdown. The less precautionary behavior was especially observed among those following unofficial sources of information after the lockdown compared to those following official government sites [79,80]. During a humanitarian crisis, such as a pandemic, the spread of rumors, and misleading information through social media and various other communication mediums, especially unofficial sources, can further impact the KAP level of community people, which leads to an increase in morbidity and mortality [81]. Such misinformation might affect the psychology and behavioral patterns of people hindering even simple practices such as hand washing, and social distancing [82]. Besides, in such crisis, the people used to practice preventive methods based on fear is not sustainable. The fear was just a result of exposure of people to numerous negative news and fear-based advertisements in different media during the rise of cases [83,84]. Therefore, it is crucial to trace the source of misinformation, improve the way of communication and make people aware of its impact for sustainable behavior change [85]. The regular monitoring and study regarding KAP on COVID-19 during the pandemic should be prioritized by concerned authorities.

Our study uncovered that the KAP on COVID-19 among the general population in the most affected districts was influenced by socio-demographic characteristics. The people living in the urban area had good knowledge of COVID-19 and good practice in prevention compared to those living in rural areas. The findings were similar to the study conducted in Bangladesh [41,42,86] and the United States [87]. This could be because that the people living in rural areas do not have easy access to information sources. In contrast, urban people have easy

access to various types of information sources that increase their familiarity on COVID-19 [88]. Besides, protection facilities are readily available in these areas to adopt preventive measures compared to rural areas.

The respondents with white-collar occupations had good KAP. The result was in line with the study conducted in Ethiopia, which showed that the people working as government officers and merchants were more likely to have sufficient knowledge of COVID-19 than those working as laborers [88]. Similar studies conducted in Turkey [43] and China [44] also illustrated the higher knowledge level on COVID-19 among white-collar workers. The respondents having white-collar occupations mostly have easy access to information, and strong learning ability further aids in increasing their attitude and practice. Our study further revealed that the highly educated respondents had good KAP on COVID-19, and this bears a resemblance to the study conducted in North East Ethiopia [39], North West Ethiopia [40], Nepal [56], Iran [48], Pakistan [52] and Bangladesh [41,42]. Highly educated people are more sensitive and better understand pertinent issues in society. Accordingly, they perceive the impact of pandemics more seriously, improving their attitude and practice for prevention along with the knowledge received [45].

Similarly, high-income respondents were observed to have good KAP in our study, recapitulating the studies conducted in Malaysia [47] and Bangladesh [42]. High-income people are mostly educated, and low-income people are least educated with lower understanding. Furthermore, females were more likely to practice preventive behavior compared to males. The finding was similar to the studies conducted in Malaysia [47], Iran [48], Indonesia [49], Cameroon [50], Uganda [51], and Bangladesh [41,42]. The online survey conducted among the respondents from 175 countries also reported a similar result [89]. This result can be supported by the fact that women are more concerned about their health because they are socialized to be more serious and responsible about themselves and their family members' health issues [49,90,91].

Knowledge itself is the foundation of learning and the basis of establishing attitude, and attitude further influence the practice of people [44]. Our study showed that the KAP differed according to the varying socio-demographic conditions. Moreover, the differences in learning abilities according to the diverse socio-demographic characteristics among the Nepalese population were also highlighted in previous literature [92,93]. The pace of obtaining knowledge, and developing the attitudes and practices of an individual depends upon their intellectual ability and prior knowledge [94]. To reduce this gap among people of different socio-demographic statuses and to make communication more effective, it is essential to identify the differences between messages disseminated and received as a result of differential exposure to the intervention, inconsistencies in interpretation, and transforming information [95]. It is important to consider the channel through which the message is disseminated, to whom the message is attributed, the response of the audience, and the feature of the message. Monitoring of the consistency of information shared through different media is also essential to avoid the confusion [96]. Therefore, our study result emphasized the need for the establishment of risk communication protocol with a clear definition of tasks among the government bodies and concerned authorities. It is indispensable to understand the people's adjustment towards recommendations provided to be safe against COVID-19. For this, establishing two-way communication platforms and community feedback mechanisms are very helpful to understand and engage communities in discussion and communication. Community health volunteers should be mobilized for door-to-door visits to make people aware.

Ministry of Health and Population, Nepal Government has prepared 'Guideline for Mobilization of COVID-19 Facilitation Group in Community Level' [97], 'Public Health Standards Regarding COVID-19 during Festivals and Celebrations' [98], 'COVID-19 Micro Contaminant Plan' [99], and 'Environmental Cleaning and Disinfection Interim Guidance in the

context of COVID-19' [100], which highlighted the prevention measures in households to the community level and also the regular monitoring of preventive behavior among the community people.

The key messages highlighted in these documents should be translated into different languages and also in understandable form by the entire population, including poor and marginalized people. Besides, the information disseminated should be based on scientific evidence to convince the people. Although the 'Guideline on the Distribution of Isolation Kits for Home Isolation Cases' [101] was prepared by Nepal Government, the distribution of kits consisting of materials for preventive measures should also be emphasized, especially among the most needful people such as those with low-economic conditions or rural people who have poor behavioral practices in prevention.

The study findings further underscored a prerequisite of targeted programs to improve the KAP on COVID-19 according to gender, occupation, income, educational background, and area of residence that will aid in controlling the risk of further waves of the pandemic in the future. Moreover, the use of social norms theory should be emphasized to accomplish positive health outcomes. Therefore, it is important to understand the approach to integrating social norms for better communication and improving the knowledge, attitudes, and practices on COVID-19 [102,103]. The needful and effective steps by concerned stakeholders from diverse disciplines will help to aware the community and augment preventive behavior.

There are some limitations of the study. It is a cross-sectional study. So it cannot provide evidence over a period of time. The study was focused only on eight districts with the higher number of cases. So, the data may not be applicable and representative for entire country. Besides, the study was conducted nearly two years ago. However, the number of cases is higher in these areas till date. The non-probability sampling method was used due to the unsuitability for conducting probability sampling during the health crisis. The implications of the study are that the findings provide timely understanding among the policymakers for revising the existing policies and recognizing and emphasizing the target population to reduce the severity of the pandemic. The relationship of culture, social norms, ethnicity and information behavior with KAP on COVID-19 among the general population was not included in this study. The future research considering these factors is crucial for detailed understanding.

## Conclusions

Findings from our study are imperative to understand the perceived knowledge, attitudes and behavior among the general population in the affected areas during the COVID-19 pandemic. The people residing in urban areas, with white-collar occupations, high-income, and high-education were good at KAP on COVID-19. In addition, females were good at practicing preventive behavior although their knowledge level was lower compared to male.

The variation in KAP on COVID-19 among the people with varying socio-economic conditions was due to the difference in the level of their understanding and available resources. The study result recommends the subsequent improvement in existing plans, policies and guidelines with a targeted approach. Moreover, the emphasis should be given to the effective implementation of the programs, accessibility of resources among all the population for the compliance of preventive measures, development of better communication strategy for their effective implementation considering effectual dissemination of information as well as infodemic management. The multifaceted approach, with the significant contribution of concerned stakeholders and their proper co-ordination including government bodies is essential to deal with the challenges during such a health crisis by effectively utilizing the available limited resources.

## Supporting information

**S1 Questionnaire.**
(DOCX)

**S1 Data.**
(XLSX)

## Acknowledgments

The authors would like to thank all the volunteers associated with Nepal Red Cross Society (NRCS) district chapters for their contribution to the questionnaire survey and valuable suggestions during the survey; and all the survey respondents.

## Author Contributions

**Conceptualization:** Rabin Malla, Sadhana Shrestha.

**Data curation:** Salina Shrestha.

**Formal analysis:** Salina Shrestha.

**Funding acquisition:** Rabin Malla.

**Investigation:** Salina Shrestha, Rabin Malla, Pallavi Singh, Jeevan B. Sherchand.

**Methodology:** Salina Shrestha, Rabin Malla, Sadhana Shrestha, Pallavi Singh.

**Project administration:** Salina Shrestha, Rabin Malla, Pallavi Singh.

**Supervision:** Salina Shrestha, Rabin Malla, Sadhana Shrestha, Pallavi Singh, Jeevan B. Sherchand.

**Writing – original draft:** Salina Shrestha, Rabin Malla.

**Writing – review & editing:** Salina Shrestha, Rabin Malla, Sadhana Shrestha, Jeevan B. Sherchand.

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
