## [Decision Letter · Decision Letter 0]

6 Mar 2023

PGPH-D-22-01721

Knowledge, Attitude and Practice (KAP) study on COVID-19 among the general population of Nepal

Dear Dr. Malla,

Thank you for submitting your manuscript to PLOS Global Public Health. After careful consideration, we feel that it has merit but does not fully meet PLOS Global Public Health’s publication criteria as it currently stands. Therefore, we invite you to submit a revised version of the manuscript that addresses the points raised during the review process.

We look forward to receiving your revised manuscript.

Kind regards,

Ismail Ayoade Odetokun, DVM, Ph.D.

Academic Editor

Journal Requirements:

1. Fig 1: please (a) provide a direct link to the base layer of the map (i.e., the country or region border shape) and ensure this is also included in the figure legend; and (b) provide a link to the terms of use / license information for the base layer image or shapefile. We cannot publish proprietary or copyrighted maps (e.g. Google Maps, Mapquest) and the terms of use for your map base layer must be compatible with our CC-BY 4.0 license. 

2. Please amend your Data Availability Statement and indicate where the data may be found

Additional Editor Comments (if provided):

Reviewers' comments:

Reviewer's Responses to Questions

**Comments to the Author**

1. Does this manuscript meet PLOS Global Public Health’s publication criteria? Is the manuscript technically sound, and do the data support the conclusions? The manuscript must describe methodologically and ethically rigorous research with conclusions that are appropriately drawn based on the data presented.

Reviewer #1: Yes

Reviewer #2: Partly

Reviewer #3: Yes

2. Has the statistical analysis been performed appropriately and rigorously?

Reviewer #1: Yes

Reviewer #2: I don't know

Reviewer #3: Yes

3. Have the authors made all data underlying the findings in their manuscript fully available (please refer to the Data Availability Statement at the start of the manuscript PDF file)?

Reviewer #1: Yes

Reviewer #2: No

Reviewer #3: No

4. Is the manuscript presented in an intelligible fashion and written in standard English?

Reviewer #1: Yes

Reviewer #2: Yes

Reviewer #3: Yes

5. Review Comments to the Author

Reviewer #1: This study is already improved by suggestions of previous reviewers. I have some minor comments.

Minor Comments

1. Write ethical approval number of the study.

2. Still this study lacks comparison with other study of Nepal. Compare your study with these study in discussion.

https://doi.org/10.1371/journal.pgph.0000737 (South Asia)

https://doi.org/10.3390/vaccines9121448 (Vaccination)

http://orcid.org/0000-0001-6218-0500

https://doi.org/10.5195/ijms.2020.669

And with some other relevant studies.

3. In figure 1, spelling of Chitwan district is written as Chitawan, correct that.

Reviewer #2: This is an important study and can add to the information on COVID-19 from Nepal. My queries and suggestions are as follows.

Introduction:

One of the challenges with a rapidly evolving disease like COVID-19 is the rapidly changing information. Will it be possible for the authors to kindly update the information in the introduction section?

Methodology:

Why did the authors confine their study to only eight districts? These are the more prosperous and developed districts of the country and the data obtained may not be applicable to the entire country.

How exactly were participants selected?

Was the sample collected representative of the population of these districts?

One of the limitations is that the data was collected nearly two years ago. The situation might be different now.

How exactly was the questionnaire developed? Was it validated and pre-tested?

The findings obtained may not be representative of the population of Nepal or of the eight districts. The authors should clearly mention that their study was carried out only in eight districts of the country in the title and in other areas of the manuscript.

The limitations section may need to mention the point about the study being done only in eight districts, and the fact that the study was conducted nearly two years ago. I am not sure about the representativeness of the sample.

Some language corrections are required but the standard of English is good.

I recommend a specialized statistical review of the manuscript.

I also recommend that the authors deposit the de-identified study data in a publicly available repository.

Reviewer #3: Nepal

Title: please modify as "Knowledge, Attitudes, and Practices on

COVID-19 among the general population of Nepal". This change should reflect throughout the manuscript.

Abstract

The abstract appears well-written but should be improved upon. For instance, authors should include the study design used in this study, the sample size, and the sampling technique used.

Keywords: The authors have not listed any keywords. Relevant keywords, as guided by the journal, should be included after the abstract.

Introduction

The authors should modify the citation style used throughout the manuscript is not in conformity with the journal's format.

L35-36: Authors should include current statistics on the number of cases and mortality of COVID-19 in Nepal.

Generally, the introduction is well-written. However, I think the authors have not highlighted why this research is important and relevant to the control of COVID-19 in Nepal. Some other studies have been done previously concerning COVID-19 in Nepal. Some of these studies are listed below. The authors should highlight within the manuscript why this study adds better or unique knowledge than the previous studies.

https://www.sciencedirect.com/science/article/pii/S2049080122016211

https://journals.sagepub.com/doi/full/10.1177/10105395211014647

https://ijms.info/IJMS/article/view/669/1111

https://www.frontiersin.org/articles/10.3389/fpubh.2022.875727/full

https://www.nepjol.info/index.php/NJST/article/view/45777

https://journals.plos.org/globalpublichealth/article?id=10.1371/journal.pgph.0000737

Methods

L96: Methodology should be replaced with "Methods"

L99: Why was the listed districts sampled in Nepal?

L129: Please include the ethical review number granted by the review council.

L157-163: The authors should provide details of the questions asked to test the attitudes and practices levels of the respondents. The sources of the questions asked should also be cited. In addition, a copy of the questionnaire should be attached as a supplementary file.

L165: Are the socioeconomic characteristics part of the main questionnaire? If yes, this should be stated.

The results and discussion sections are well presented. However, the authors should do more by comparing the results obtained with other relevant studies in and around Nepal.

The authors should remove the yellow highlights used throughout the manuscript. Do these highlights mean something?

Track changes should also be removed.

6. PLOS authors have the option to publish the peer review history of their article (what does this mean?). If published, this will include your full peer review and any attached files.

**Do you want your identity to be public for this peer review?** For information about this choice, including consent withdrawal, please see our Privacy Policy.

Reviewer #1: No

Reviewer #2: **Yes: **Pathiyil Ravi Shankar

Reviewer #3: No

---

## [Decision Letter · Decision Letter 1]

14 Jun 2023

Knowledge, attitudes and practices (KAP) on COVID-19 among the general population in most affected districts of Nepal

PGPH-D-22-01721R1

Dear Dr. Malla,

We are pleased to inform you that your manuscript 'Knowledge, attitudes and practices (KAP) on COVID-19 among the general population in most affected districts of Nepal' has been provisionally accepted for publication in PLOS Global Public Health.

Best regards,

Ismail Ayoade Odetokun, DVM, Ph.D.

Academic Editor

Reviewer Comments (if any, and for reference):

Reviewer's Responses to Questions

**Comments to the Author**

1. If the authors have adequately addressed your comments raised in a previous round of review and you feel that this manuscript is now acceptable for publication, you may indicate that here to bypass the “Comments to the Author” section, enter your conflict of interest statement in the “Confidential to Editor” section, and submit your "Accept" recommendation.

Reviewer #2: All comments have been addressed

Reviewer #3: All comments have been addressed

2. Does this manuscript meet PLOS Global Public Health’s publication criteria? Is the manuscript technically sound, and do the data support the conclusions? The manuscript must describe methodologically and ethically rigorous research with conclusions that are appropriately drawn based on the data presented.

Reviewer #2: Yes

Reviewer #3: Yes

3. Has the statistical analysis been performed appropriately and rigorously?

Reviewer #2: Yes

Reviewer #3: Yes

4. Have the authors made all data underlying the findings in their manuscript fully available (please refer to the Data Availability Statement at the start of the manuscript PDF file)?

Reviewer #2: Yes

Reviewer #3: Yes

5. Is the manuscript presented in an intelligible fashion and written in standard English?

Reviewer #2: Yes

Reviewer #3: Yes

6. Review Comments to the Author

Reviewer #2: The authors have addressed the comments and the manuscript can be accepted for publication.

Reviewer #3: (No Response)

7. PLOS authors have the option to publish the peer review history of their article (what does this mean?). If published, this will include your full peer review and any attached files.

**Do you want your identity to be public for this peer review?** For information about this choice, including consent withdrawal, please see our Privacy Policy.

Reviewer #2: **Yes: **Pathiyil Ravi Shankar

Reviewer #3: No
